# Insights into the ISG15 transfer cascade by the UBE1L activating enzyme

Iona Wallace [1], Kheewoong Baek [2], J. Rajan Prabu [2], Ronnald Vollrath[2], Susanne von Gronau[2], Brenda A. Schulman [2] ✉ & Kirby N. Swatek [1,2] ✉

The attachment of the ubiquitin-like protein ISG15 to substrates by specific E1-E2-E3 enzymes is a well-established signalling mechanism of the innate immune response. Here, we present a 3.45 Å cryo-EM structure of a chemically trapped UBE1L-UBE2L6 complex bound to activated ISG15. This structure reveals the details of the first steps of ISG15 recognition and UBE2L6 recruitment by UBE1L (also known as UBA7). Taking advantage of viral effector proteins from severe acute respiratory coronavirus 2 (SARS-CoV-2) and influenza B virus (IBV), we validate the structure and confirm the importance of the ISG15 C-terminal ubiquitin-like domain in the adenylation reaction. Moreover, biochemical characterization of the UBE1L-ISG15 and UBE1L-UBE2L6 interactions enables the design of ISG15 and UBE2L6 mutants with altered selectively for the ISG15 and ubiquitin conjugation pathways. Together, our study helps to define the molecular basis of these interactions and the specificity determinants that ensure the fidelity of ISG15 signalling during the antiviral response.

The antiviral innate immune response is initiated by danger sensors, termed pattern recognition receptors (PRRs), which first recognize viral pathogen-associated molecular patterns (PAMPs) and subsequently activate downstream signaling pathways[1]. In phase one of this defence response, the interferon regulatory factors (IRF-3 and IRF-7) induce the expression of interferons (IFNs) α and β. Once outside of the cell, IFNs function as autocrine and paracrine cytokines and activate cell surface receptors surrounding the infection site[2]. In phase two of this response, the activated interferon receptors signal through the Janus kinase (JAK)-signal transducer and activator of transcription (STAT) pathway to assemble and translocate the IFN-stimulated gene factor 3 (ISGF3) to the nucleus[3]. The outcome of ISGF3 transcriptional activation is the expression of hundreds of interferon-stimulated genes (ISGs) and establishment of the antiviral state in cells.

ISGs antagonize viral infection through multiple discrete mechanisms. Cytokine and chemokine secretion, expression of antiviral effectors, enhanced pathogen surveillance, and widespread remodeling of the proteome, all contribute to the IFN response and hence create a highly efficient barrier against many viruses[4]. Previous research has uncovered the function of individual ISGs, however the

antiviral state is the concerted action of multiple ISGs and global rearrangements to the host proteome, which includes rapid and sustained alterations to the landscape of post-translational modifications.

Ubiquitin and ubiquitin-like modifications play a central role in this defence response. PRR signaling and NF-κB transcriptional activation depend on ubiquitin signaling. Moreover, several ISGs encode proteins that assemble and disassemble these modifications. The ubiquitin-like protein, interferon-stimulated gene 15 (ISG15), is one of the most abundant ISGs[5,6] and through a highly specific E1-E2-E3 enzyme cascade modifies hundreds of proteins during the IFN response. Conversely, these modifications can be removed by deISGylases, including USP18[7,8]. In addition to conjugation-dependent functions, free (unconjugated) ISG15 acts extracellularly as a cytokine to stimulate IFN-γ secretion and intracellularly to suppress JAK-STAT signaling through a non-catalytic interaction with USP18[9–12].

These functional roles for free ISG15 were uncovered while studying ISG15 deficient patients, who interestingly displayed a lack of viral disease, but increased susceptibility to mycobacterial disease and an enhanced type-I interferon response[10,12]. The hyperactive interferon response in ISG15 deficient patients is similarly observed in USP18

[1]Medical Research Council Protein Phosphorylation and Ubiquitylation Unit, School of Life Sciences, University of Dundee, Dundee DD1 5EH, UK. [2]Department of Molecular Machines and Signaling, Max Planck Institute of Biochemistry, Am Klopferspitz 18, 82152 Martinsried, Germany. ✉e-mail: schulman@biochem.mpg.de; kswatek001@dundee.ac.uk

deficient patients[13]. In contrast, ISG15 knock-out mice are more susceptible to numerous viruses[9,14]. Together, these findings underscore species-specific differences in the ISG15 pathway and the importance of understanding the regulatory mechanisms of different forms of ISG15 during innate immune signaling.

Reminiscent of linear diubiquitin, ISG15 is composed of two ubiquitin-like domains fused together through a short linker sequence. The N-terminal and C-terminal domains share very little sequence conservation to ubiquitin (27% and 37%, respectively), however the last six C-terminal residues – the point of substrate attachment – are entirely conserved with ubiquitin. The enzymes responsible for ISG15 conjugation were first described over 15 years ago and include: the E1 activating enzyme UBE1L/UBA7[15], the E2 conjugating enzyme UBE2L6/UbcH8[16], and the HECT E3 ligase HERC5[17]. In a multi-step process, ISG15 is first activated or adenylated by UBE1L, then sequentially transferred to the catalytic cysteines of the E1-E2-E3 enzymes, and targeted to substrate lysine residues.

Here, we present a cryo-EM structure of the initial catalytic steps of the ISG15 pathway. Using a chemical biology approach which allows the capture of these transient reaction intermediates[18,19], we cross-link the active sites of UBE1L and UBE2L6 to form a multiprotein complex assembly. This approach captures the ISG15 adenylate intermediate and reveals the structural features required for ISG15 and UBE2L6 recognition. A similar strategy was recently reported to solve another UBE1L-ISG15-UBE2L6 structure while this paper was under review[20]. In addition to the structures, biochemical analysis using viral effector proteins and site-directed mutagenesis probe the mechanistic basis of this specificity. These analyses enable the design of ISG15 and UBE2L6 mutants with altered selectively for the ISG15 and ubiquitin pathways. We anticipate the principles of ISG15 recognition and transfer described here will apply to many ISG15-containing species, as well as to the downstream steps of the ISG15 pathway, and therefore this study sheds light on this important, yet enigmatic post-translational modification.

## Results

### Assembly of UBE1L complexes

To visualize the first step of the ISG15 cascade (Fig. 1a), we trapped UBE1L (Cys599Ala) in complex with Mg-ATP and ISG15 (Supplementary Fig. 1a, b). Cryo-EM imaging produced clear two-dimensional (2D) and three-dimensional (3D) classes, which allowed 3D reconstruction (Supplementary Fig. 1c, d). Secondary structures were not visible, however the model resembled previously described E1 structures[21–26] and therefore allowed for the placement of individual domains. The UBE1L catalytic cysteine domain (i.e. second catalytic cysteine half-domain, SCCH) was noticeably absent in the structure, likely due to its flexibility (Supplementary Fig. 1d, e). The lack of high-resolution details limited structural interrogation of the complex.

In order to obtain a stable structural intermediate with high resolution information, we assembled a ternary complex consisting of UBE1L, UBE2L6, and full-length ISG15 (Fig. 1b, c). As previously described, E2 enzymes are coordinated through multiple E1 domains[18,22,27–29], including the SCCH domain (discussed in detail below). We therefore rationalized that stabilization of UBE1L domains via UBE2L6 binding might facilitate cryo-EM analysis, while simultaneously providing insight into the mechanism of ISG15 transfer from E1 to E2. To assemble the E1-E2 ISG15 complex, we first cross-linked the active site cysteines of UBE1L and UBE2L6 by adapting a method from Lima and colleagues for studying other E1-E2 pathways[18]. To prevent non-specific crosslinking, the non-catalytic UBE2L6 cysteine residues were mutated to serine (UBE2L6^{C86only}; Supplementary Fig. 2a), which was also utilized for UBE2L6 in a recent publication by Olsen and colleagues[20]. Importantly, UBE2L6^{C86only} retained its ability to form a UBE2L6 - ISG15 thioester bond with fluorescent ISG15 (Supplementary Fig. 2b, c) and functioned with the ISGylation machinery in cells (Supplementary Fig. 2d), underscoring the utility of this mutant for structural analysis. The disulfide-linked UBE1L-UBE2L6 was then incubated with Mg-ATP and ISG15 and purified by analytical size-exclusion

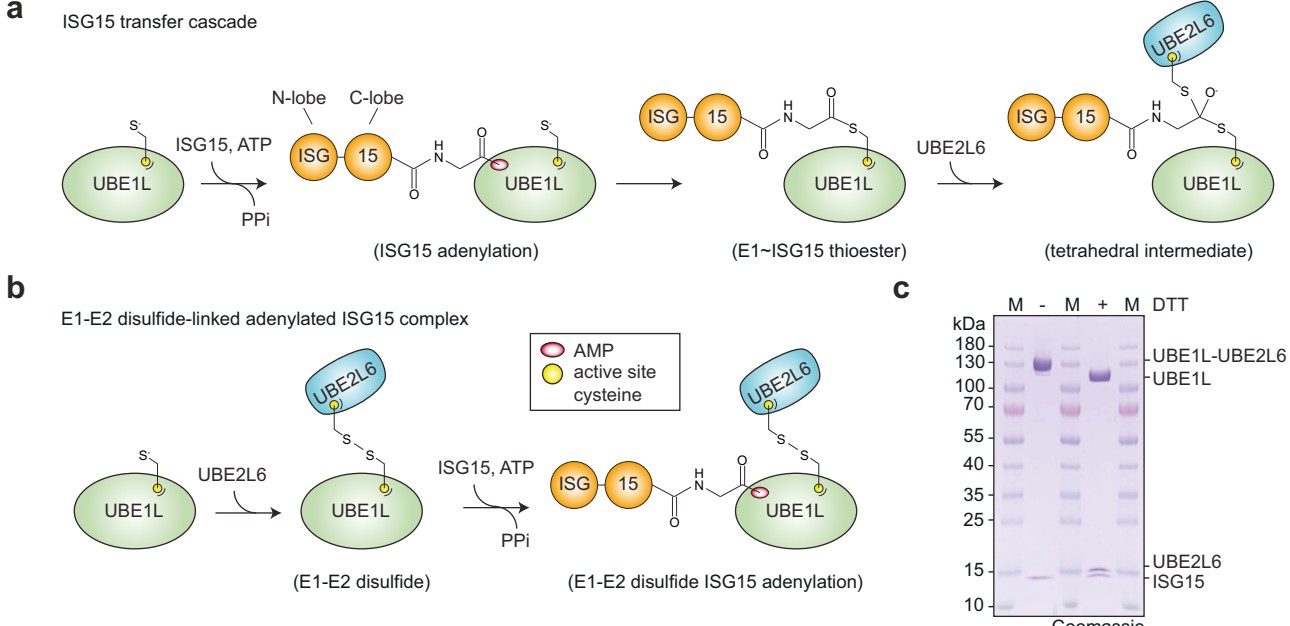

**Fig. 1 | The ISG15 transfer cascade and ISG15 E1-E2 complex formation.**
**a** Schematic of ISG15 adenylation and transfer through the E1 activating enzyme UBE1L to the E2 conjugating enzyme UBE2L6. In an ATP-dependent manner, UBE1L adenylates or 'activates' the C-terminus of ISG15. This high-energy ISG15 intermediate reacts with the catalytic cysteine of UBE1L forming a thioester bond (- denotes thioester bond). Subsequently, ISG15 is transferred to UBE2L6 forming a tetrahedral intermediate. **b** Schematic of ISG15 E1-E2 disulfide-linked complex formation. The active site cysteines of UBE1L and UBE2L6 were initially cross-linked

via a disulfide bond (- denotes disulfide bond). Subsequently, this E1-E2 disulfide-linked complex was incubated with ISG15 and Mg-ATP to form a ternary protein complex with adenylated ISG15. **c** SDS-PAGE of the purified E1-E2 ISG15 adenylated complex. The addition of dithiothreitol (DTT) to the complex reduced the disulfide bond between UBE1L and UBE2L6. Formation of the E1-E2 ISG15 adenylated complex was performed independently in duplicate. Source data are provided in the Source Data file.

chromatography (Supplementary Fig. 2e). The application of reducing agent to the complex hydrolyzed the E1-E2 disulfide linkage and confirmed stoichiometric complex formation (Fig. 1c, Supplementary Fig. 2e).

**Cryo-EM structure of UBE1L in complex with ISG15 and UBE2L6**
Using cryo-EM, we obtained a 3.45 Å structure of the UBE1L-UBE2L6-ISG15 complex (Fig. 2a, Supplementary Fig. 3, Supplementary Table 1). Similar to other E1 enzymes[30], and as seen in a related study published while our paper was in revision[20], the UBE1L structure consists of four distinct domains: the adenylation domain (AD; which contains both the inactive adenylation domain & active adenylation domain), first catalytic cysteine half domain (FCCH), second catalytic cysteine half domain (SCCH) and ubiquitin-fold domain (UFD) (Fig. 2a, Supplementary Fig. 1e). The adenylation domain, which activates ISG15 via ATP-dependent adenylation of the C-terminal glycine, forms the largest contact surface with ISG15. Following adenylation, ISG15 forms a thioester bond with the SCCH domain. As anticipated, the active site cysteines of the SCCH domain and UBE2L6 were located within proximity for disulfide bond formation. The last domain, the UFD, adopts a well-characterized binding mode for E1-mediated E2 recruitment[28], where the UFD engages the N-terminal helix of UBE2L6. ISG15 is located at the center of the complex, with multiple UBE1L domains cradling the C-terminal ubiquitin-like fold (C-lobe, Fig. 2b).

Interestingly, the N-terminal ubiquitin-like fold (N-lobe) is not visible in the cryo-EM density, presumably due to its flexibility during the adenylation reaction. At the top of the complex rests UBE2L6 which, as expected, is sandwiched between multiple UBE1L domains (Fig. 2c). The UFD and SCCH domains grasp opposite sides of the E2-fold, and the crossover loop, which connects the adenylation and SCCH domains, is located between UBE2L6 and ISG15 (Fig. 2b, c).

Since the ISG15 N-lobe was not visible in our initial structure, we were curious if further processing of the cryo-EM images could resolve this domain. Indeed, our reanalysis identified a particle population which contained density surrounding the N-lobe (Supplementary Fig. 4, Supplementary Table 1). Apart from the presence of the ISG15 N-lobe, the overall reconstruction resembled the high-resolution UBE1L-UBE2L6 ISG15 structure. In this subclass of particles, the N-lobe protrudes outward from the complex and does not make substantial contact with UBE1L (Fig. 2d). In contrast, a recent study suggests the ISG15 N-lobe contacts the UFD[20]. Together these results highlight the flexibility of the N-lobe during the adenylation reaction, which is further supported by the low resolution of this domain in both structures (Supplementary Fig. 4b).

**ISG15 and UBE2L6 recognition by UBE1L**
To understand the molecular basis of UBE1L's specificity for ISG15 and UBE2L6, we compared the domain architecture and key interactions of

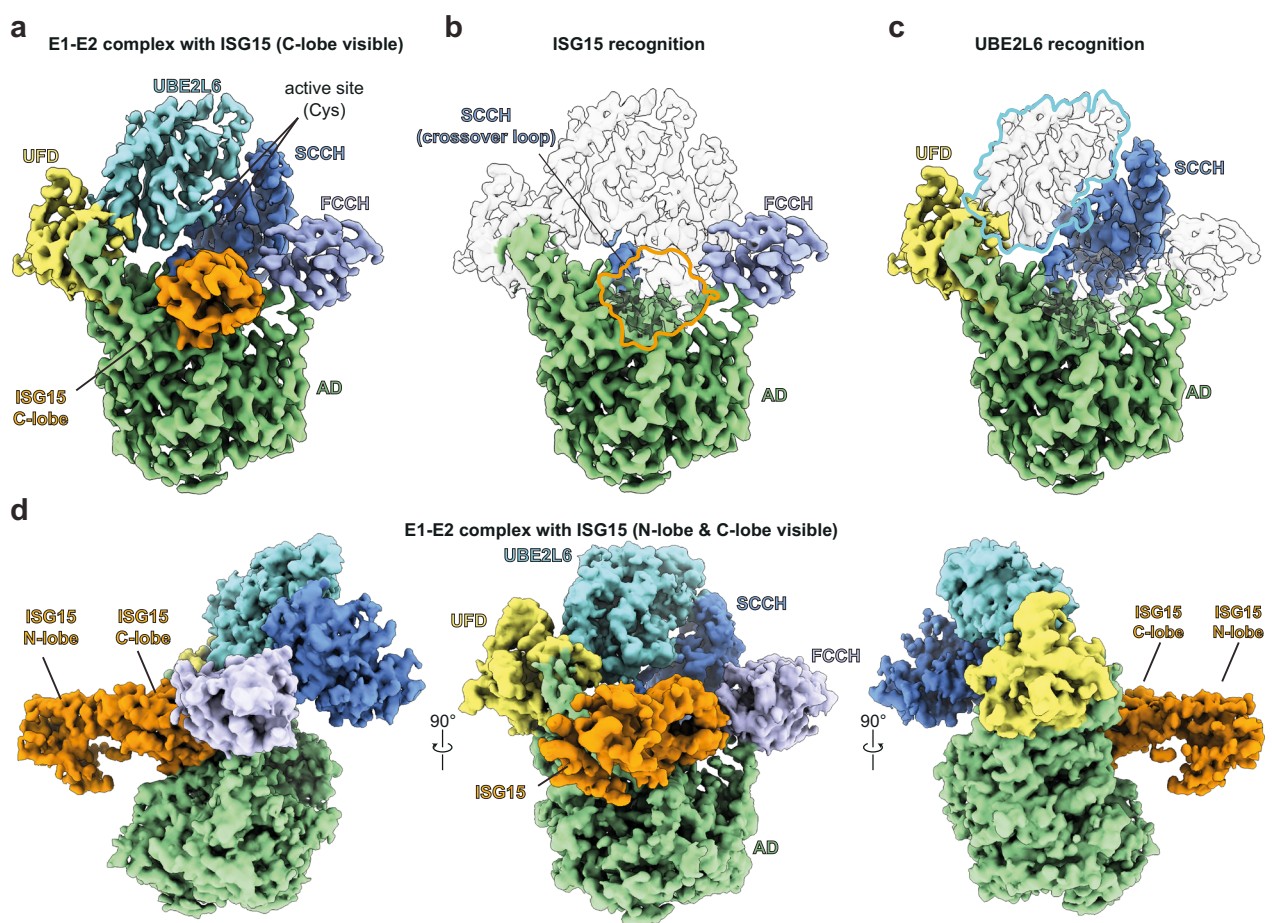

**Fig. 2 | Cryo-EM structure of ISG15 in complex with UBE1L and UBE2L6. a** Cryo-EM density of UBE1L in complex with UBE2L6 and ISG15. For ISG15, the density surrounding the C-terminal ubiquitin-like fold (C-lobe) is visible. **b** The UBE1L domains contacting ISG15 are highlighted and include the adenylation domain (AD), first catalytic cysteine half-domain (FCCH), and cross-over loop of the second catalytic cysteine half-domain (SCCH). As expected, ISG15 is positioned within the adenylation catalytic module. **c** The UBE1L domains contacting UBE2L6 are highlighted and include the UFD (ubiquitin fold domain) and SCCH. UBE2L6 is located at the top of the structure and the E1-E2 catalytic cysteines are positioned for disulfide bond formation. **d** Cryo-EM density of a subclass of the complex (DeepEMhancer map). The N-terminal ubiquitin-like fold of ISG15 (N-lobe) is visible in the density, however the N-lobe does not make significant contact with UBE1L.

our structure to other ubiquitin and ubiquitin-like protein E1 enzyme structures[21,22,24,25]. Overall, ubiquitin and ubiquitin-like proteins adopt similar positions during the adenylation reaction (Supplementary Fig. 5a–e). Interestingly, we observed density extending from ISG15's C-terminus into the adenylation active site, indicating the ISG15 C-terminal glycine is adenylated in our structure (Fig. 3a). Comparison of our structure to a ubiquitin E1 structure containing an adenylated ubiquitin C-terminus[24] revealed similar active sites and AMP orientations (Supplementary Fig. 5f), highlighting the conservation of the E1 adenylation reaction.

Next, we examined the interactions between UBE1L and ISG15 to understand the mechanism of ISG15 specificity. The adenylation domain, FCCH domain, and crossover loop of UBE1L contact three distinct surfaces of ISG15 (Fig. 2b). The adenylation domain formed the largest interaction surface, and comparison to the Uba1 ubiquitin structure revealed unique ISG15 interaction sites (Fig. 3b, c; Supplementary Fig. 5g). Both adenylation domains contain a hydrophobic β-sheet which contacts the Ile44 patch of ubiquitin (Ile44/His68/Val70) and structurally equivalent patch in ISG15 (i.e. Thr125/F149/N151). While differences between these interaction sites exist, more obvious differences surround Trp123 of ISG15, a well-known recognition site for deISGylases[31–36]. In particular, ISG15's Trp123 and P130 create a unique hydrophobic patch which forms additional interactions with the hydrophobic β-sheet of UBE1L (e.g., Tyr885 and Tyr896; Fig. 3c; Supplementary Fig. 5g). Importantly, these hydrophobic contacts do not exist in the Uba1-ubiquitin complex but are instead replaced with an ionic interaction (Supplementary Fig. 5g).

The overall position of UBE2L6 resembles previously characterized E1-E2 structures and therefore UBE1L's specificity for UBE2L6 is likely the result of unique binding surfaces rather than large structural differences (Fig. 3d, Supplementary Fig. 6a–d)[18,29]. Modeling of a phylogenetically related ubiquitin E2, UBE2L3 (sequence similarity of 71.4%), onto UBE2L6 helped provide insight into this specificity (Supplementary Fig. 6b, e). In the α1 helix of UBE2L6, Met5 and Val8 form hydrophobic contacts with the UFD (Fig. 3e, Supplementary Fig. 6e). Conversely, in UBE2L3 the analogous residues (Arg5, Met8) would form a less productive interaction interface (Supplementary Fig. 6e). Another potential interesting contact site surrounds Met123 of UBE2L6, which based on cryo-EM density contacts the SCCH domain (Supplementary Fig. 6f). The equivalent position in UBE2L3 is occupied by a less hydrophobic alanine residue and therefore this UBE2L6 residue might influence UBE1L binding. Overall, our analysis of the key ISG15 and UBE2L6 contacts with UBE1L is broadly similar to that made by a recent study[20].

## Altering ISG15 pathway specificity

To further explore the specificity determinants of the ISG15 enzyme cascade, we set out to validate our structural analysis using biochemical approaches. Visualization of ISG15 residues contacting UBE1L, comprising a 'patch' analogous to ubiquitin's Ile44 patch, revealed the surface features important for E1 recognition (Fig. 4a, Supplementary Fig. 7a). Based on these contact sites, we designed and purified ISG15 truncations, domain substitutions, and site-specific mutations to test the effect of these mutants in UBE1L charging assays (Fig. 4b). Consistent with the structure, deletion of the ISG15 N-lobe did not alter the efficiency of UBE1L charging (Fig. 4c). To help rule out the possibility of subtle differences in E1 ~ ISG15 thioester formation, we measured the ATP dependency of UBE1L charging (Supplementary Fig. 7b). Even in conditions with low levels of ISG15 and ATP, E1 thioester formation with full-length and C-lobe-only ISG15 was comparable (Supplementary Fig. 7c), suggesting that residues within the C-lobe are the main determinants of UBE1L's ISG15 selection. Moreover, swapping the ISG15 N-lobe with either ubiquitin or SUMO1 – a Ub/SUMO1-ISG15 C-lobe fusion – had no impact on UBE1L charging (Fig. 4d, Supplementary Fig. 7d), further validating that the ISG15 N-lobe is dispensable for E1 charging.

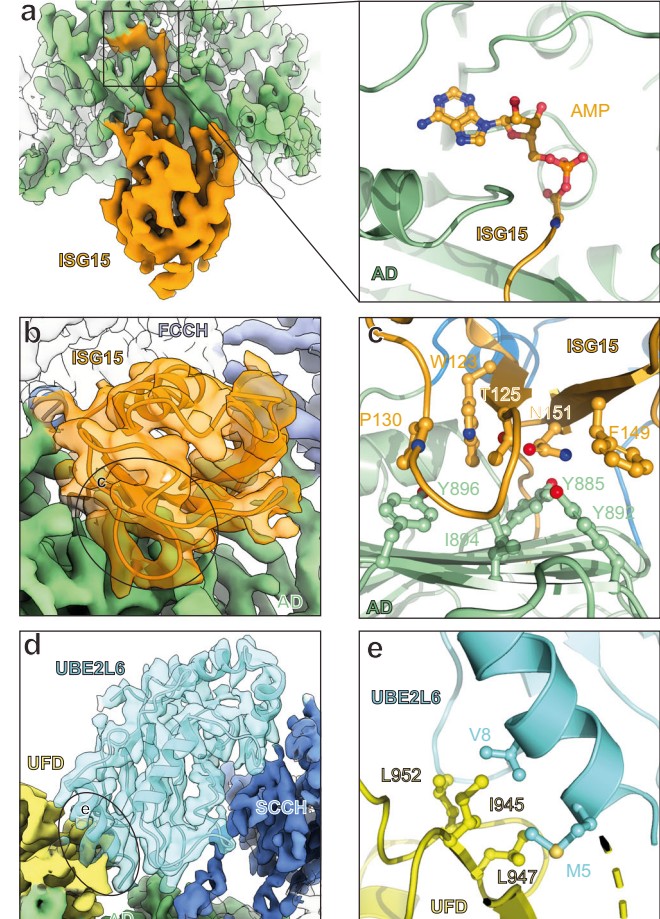

**Fig. 3 | Structural analysis of UBE1L specificity. a** Left, cryo-EM density of adenylated ISG15 bound to UBE1L. Right, close-up view of the adenylated ISG15 C-terminus within the adenylation domain (AD) active site (also see Supplementary Fig. 5f). **b** UBE1L-ISG15 surface contacts. The circled area corresponds to the primary contact site between ISG15 and UBE1L. Amino acid side chain interactions within the interface are shown in **c. c** ISG15 and AD interactions. The Thr125 patch of ISG15 (Thr125, Phe149, Asn151 – analogous residues to the Ile44 patch of ubiquitin; also see Supplementary Fig. 5g) contact residues within the AD (Tyr885, Tyr892, Ile894). Additional hydrophobic residues of ISG15 (Trp123, Pro130) contact UBE1L (Tyr896). **d** UBE1L recognition of UBE2L6. Circled area corresponds to the primary contact site between the ubiquitin fold domain (UFD) of UBE1L and UBE2L6. The second catalytic cysteine half-domain (SCCH) of UBE1L also contacts UBE2L6. Amino acid side chain interactions within the UFD-UBE2L6 interface are shown in **e. e** UBE2L6 and UFD interactions. A hydrophobic surface of the UFD (Ile945, Leu947, Leu952) coordinates UBE2L6 helix-1 residues (Met5, Val8).

Since ubiquitin and ISG15 function selectively with their cognate E1-E2-E3 enzymes, we wondered whether swapping the ISG15 residues contacting UBE1L with the analogous residues of ubiquitin would unmask the determinants of their selectivity (Fig. 4a). Interestingly, a heavily 'ubiquitylized' ISG15 mutant (W123R/T125I/P130Q/F149H/N151V or ISG15$^{5xmut}$), in which the entire ISG15-adenylation domain interface was replaced, had marginal effects on UBE1L charging, but strikingly led to complete mis-activation and thioester formation with the ubiquitin E1 enzyme, UBA1 (Fig. 4e–g, Supplementary Fig. 7e–f). Consistent with the ISG15$^{5xmut}$, an ISG15$^{4xmut}$ mutant (T125I/P130Q/F149H/N151V) displayed similar levels of UBA1 charging (Fig. 4f). Conversely, an ISG15$^{3xmut}$ mutant (T125I/F149H/N151V), which lacked the P130Q mutation, behave similarly to wild-type ISG15 (Fig. 4f). However, the ISG15 P130Q mutant did not confer the same gain-of-function activity. Together, these results highlight the importance of ISG15 'patches' in E1 recognition.

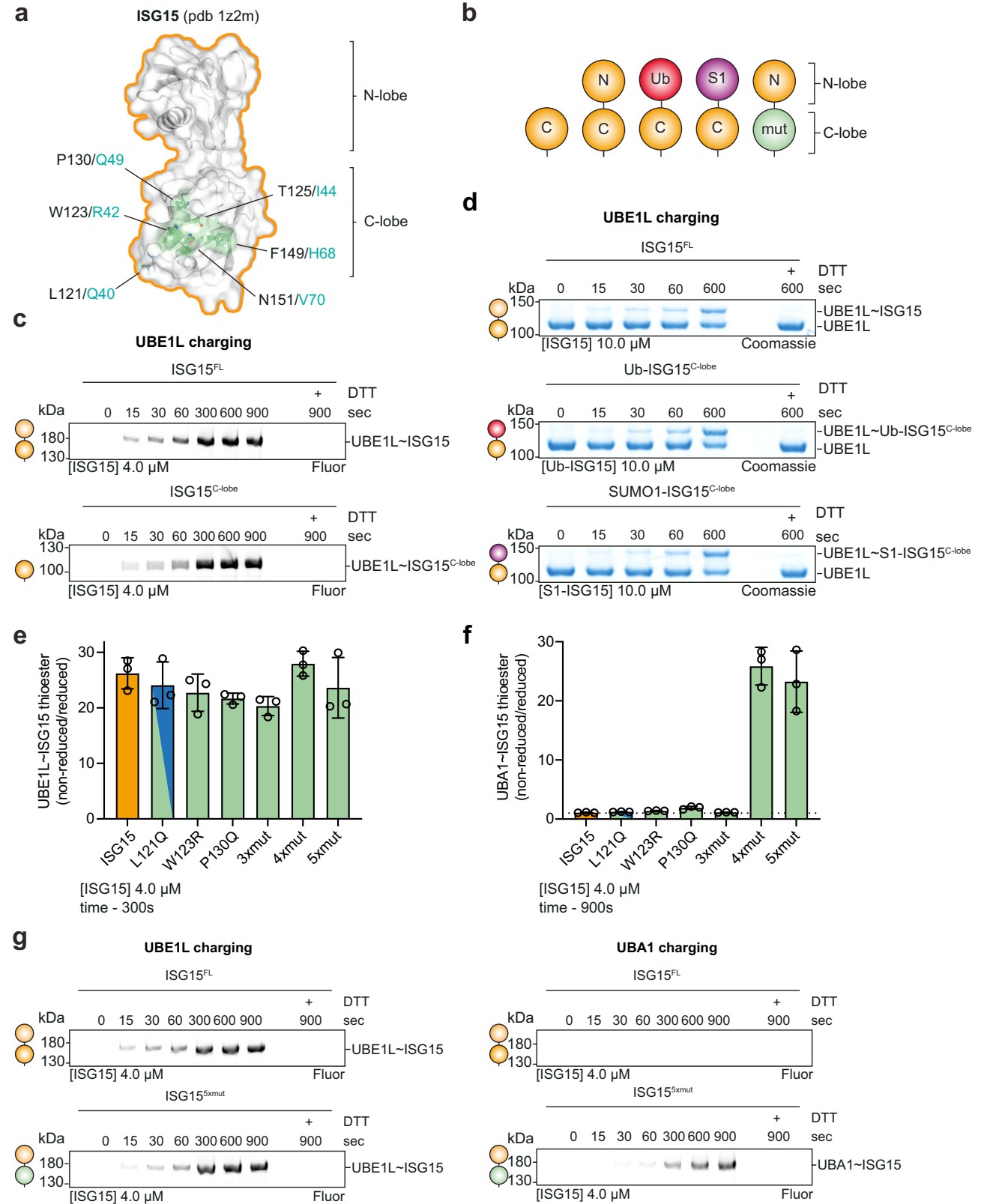

Expanding upon our observation that the ISG15$^{4xmut}$ and ISG15$^{5xmut}$ mutants are capable of UBA1-mediated mis-activation, we tested if this mis-activation also applied to ubiquitin E2 conjugating enzymes. Previous work has shown that E1 interactions substantially determine an E2's selectivly[27,37,38]; therefore, we reasoned that these ISG15 mutants might form thioester conjugates with ubiquitin E2 enzymes. Comparison of wild-type ISG15 and ISG15$^{5xmut}$, revealed slight differences in

UBE1L-mediated UBE2L6 charging (Fig. 5a). However, comparison of UBA1-mediated UBE2L6 charging to the phylogenetically related ubiquitin E2 enzyme UBE2L3, revealed that the ISG15$^{5xmut}$ forms a thioester linkage with UBE2L3 (Fig. 5b). Remarkably, UBA1-mediated UBE2L3 charging with ISG15$^{5xmut}$ was comparable to UBE1L-mediated UBE2L6 charging with wild-type ISG15 and ISG15$^{5xmut}$ in our assays, although it was reduced compared to UBA1-mediated UBE2L3 charging with

**Fig. 4 | Biochemical characterization of ISG15 specificity. a** Structure of ISG15 (pdb 1z2m) highlighting residues that contact UBE1L. Residues are shown in ball-and-stick representation under a semi-transparent surface. Interactions are exclusively located in the C-terminal ubiquitin-like fold (ISG15 C-lobe; also see Fig. 3). Analogous residues of ubiquitin are shown in teal. **b** Diagram representing ISG15 constructs used in **c–g**. From left to right: ISG15^C-lobe, full-length ISG15 (ISG15^FL), ubiquitin(Ub)-ISG15^C-lobe fusion, SUMO1-ISG15^C-lobe fusion, ISG15 mutants (ISG15^mut(s)). **c** UBE1L charging reactions with fluorescent ISG15 and ISG15^C-lobe. Reactions were separated by SDS-PAGE and visualized with fluorescent imaging. **d** UBE1L charging reactions with ISG15^FL, Ub-ISG15^C-lobe, and SUMO1-ISG15^C-lobe. Reactions were separated by SDS-PAGE and visualized with Coomassie stain. **e** Quantification of UBE1L charging reactions with fluorescent ISG15 and ISG15 mutants. ISG15 mutants include: ISG15^3xmut (T125I/ F149H/N151V), ISG15^4xmut (T125I/P130Q/F149H/N151V), ISG15^5xmut (W123R/T125I/P130Q/F149H/N151V). Mutated ISG15 residues were swapped with the analogous residues of ubiquitin (also see Supplementary Fig. 5g). Reactions were performed as in **c**. Error values represent s.d. from the mean ($n = 3$ independent experiments). Samples derive from same experiment and gels were processed in parallel. **f** Quantification of UBA1 charging with ISG15 mutants as in **e**. Error values represent s.d. from the mean ($n = 3$ independent experiments). Samples derive from same experiment and gels were processed in parallel. **g** UBE1L and UBA1 charging reactions with fluorescent ISG15^FL and ISG15^5xmut as in **c**. Experiments were performed independently in triplicate. Source data are provided in the Source Data file.

ubiquitin (Fig. 5a, b). A systematic analysis of E2 - ISG15 charging using a panel of 28 E2 enzymes revealed that in UBA1-mediated E2 charging assays all the ubiquitin E2s tested formed a thioester conjugate with ISG15^5xmut (Fig. 5c, Supplementary Fig. 8). Conversely, the Ubl-specific E2 enzymes (UBE2I – SUMO; UBE2F & UBE2M – NEDD8; UBE2L6 – ISG15) lacked this activity (Fig. 5c).

Given the ability of the ISG15^5xmut to function with ubiquitin E1 and E2 enzymes in vitro, we hypothesized that this ISG15 mutant might similarly function with the ubiquitin machinery in cells. Indeed, transfection of the FLAG-tagged ISG15^5xmut in unstimulated HeLa cells resulted in visible ISGylation, and these modified substrates further accumulated upon proteasomal inhibition with MG132 (Fig. 6a). Importantly, transfection of wild-type ISG15 or a non-conjugatable version of ISG15^5xmut (ISG15^5xmut ΔGG) did not result in substrate ISGylation, confirming the signal observed in the ISG15^5xmut transfection assays was not a result of over-expression or the modification of ISG15^5xmut itself (Fig. 6a). Furthermore, treatment of ISG15^5xmut transfected cells with the ubiquitin E1 enzyme inhibitor, TAK-243, interfered with ISGylation, confirming the ubiquitination machinery was responsible for these modifications (Fig. 6b). Lastly, we wondered if the ISG15^5xmut can function with the canonical ISGylation machinery in cells, and showed that when transfected with the ISG15 enzyme cascade, ISG15^5xmut can modify substrates in a UBE1L-dependent manner (Supplementary Fig. 9).

**Analysis of UBE2L6 specificity**

To ascertain the importance of the E1-E2 contacts in the disulfide-linked structure, we made specific point mutations in UBE2L6 and monitored the ability of these mutants to form E2 - ISG15 conjugates. Similar to E1 thioester formation, deletion of the ISG15 N-lobe had no impact on E2 charging (Supplementary Fig. 10a). Inspection of the UBE2L6 residues contacting UBE1L helped reveal the E2 residues important for E1 binding (Supplementary Fig. 10b, c). Mutations were based on structurally equivalent residues in UBE2L3 (Supplementary Fig. 10b), which compared to UBE2L6 has low ISG15 charging activity (Supplementary Fig. 8a)[16,38]. The interaction between UBE2L6 and UFD was critical for ISG15 thioester formation (Supplementary Fig. 10d). In particular, a UBE2L6 mutant containing UBE2L3 residues at this interface (UBE2L6^M5R/V8M) significantly reduced E2 - ISG15 thioester formation, which is consistent with a previous report[38]. However, mutation of the UBE2L6-SCCH interface (i.e Met123Ala; Supplementary Fig. 6f) had minimal impact on charging (Supplementary Fig. 10d).

To determine if the UBE2L6 mutants with low levels of E2 - ISG15 activity conversely had increased activity with the ubiquitin conjugation machinery, we tested these mutants in E2-Ub charging assays and E3-mediated ubiquitin assembly reactions. Compared to wild-type UBE2L6, these mutants displayed a slight increase in activity, thereby confirming these residues are also important for UBA1-UBE2L3 recognition (Supplementary Fig. 10e, f).

**Using viral ISG15 binders as tools to study UBE1L charging**

In an effort to antagonize ISG15 signaling, many viruses have evolved strategies to remove or redirect these modifications[9,39]. For example,

both influenza B virus (IBV) and severe acute respiratory coronavirus 2 (SARS-CoV-2) encode proteins that bind ISG15, however their strategies of immune evasion are unique. SARS-CoV-2 contains a papain-like protease PLpro, which hydrolyses ISG15 modified substrates[31,33,40], while the non-structural protein 1 of influenza B virus (NS1B) was reported to bind ISG15 and inhibit UBE1L's activity in cell transfection assays[15]. Contrary to this result and more recently, it was shown that NS1B does not inhibit ISG15 conjugation during IBV infection, but rather sequesters ISGylated substrates[41].

To gain further insight into the mechanisms of ISG15 recognition by UBE1L, we used NS1B and inactive PLpro as tools to study UBE1L charging. Modeling of NS1B onto the UBE1L cryo-EM structure predicted that the interaction of NS1B with the ISG15 N-lobe would not interfere with the adenylation reaction (Fig. 7a). However, PLpro, which contacts both the ISG15 N-lobe and C-lobe, was predicted to inhibit this reaction (Fig. 7b). In agreement with our models, NS1B had no effect in UBE1L charging assays. Conversely, catalytically inactive PLpro significantly reduced ISG15 charging of UBE1L (Fig. 7c).

These results were further corroborated using fluorescence polarization (FP) assays[42], which can measure both ISG15 binding and subsequent UBE1L charging. While both NS1B and PLpro bound ISG15, as demonstrated by elevated FP signal, the UBE1L charging reactions produced different outcomes. The addition of UBE1L to the NS1B-ISG15 sample resulted in an overall higher FP signal compared to the minus NS1B control, indicating NS1B remains bound to the N-lobe of ISG15 after UBE1L charging and does not interfere with the charging reaction (Fig. 7d, e). However, the addition of UBE1L to the PLpro-ISG15 sample initially led to an increase in FP signal, but gradually returned to levels seen in the control sample, indicating a competition between inactive PLpro and UBE1L for ISG15 occupancy and a delay in UBE1L charging (Fig. 7d, e). As a control, an NS1B mutant with a defective ISG15 binding site (W36A/Q37A[43]) produced no observable increase in FP signal upon co-incubation with ISG15 or the addition of UBE1L (Fig. 7d, e). Together, these results are consistent with our structural models and further validate the importance of the ISG15 C-lobe for E1 charging.

## Discussion

Despite the realization that IFN-stimulated cells contain hundreds of proteins with ISG15 modifications[44], the mechanisms of ISG15 recognition and transfer through the E1-E2-E3 cascade have remained largely unknown. This study, and another published while ours was under revision[20], have provided structural insights into this process. First, the ISG15 C-lobe forms an intricate network of side chain interactions with the adenylation domain of UBE1L. Second, the N-lobe is dispensable for E1 and E2 thioester formation, in agreement with a previous report[45]. In addition, we discovered that the ISG15 Thr125 patch negatively selects against mis-activation through UBA1, while structurally equivalent residues of ubiquitin positively select for UBA1-mediated adenylation, which is in agreement with previous reports that show Ubl specificity is driven by negative rather than positive selection against the ubiquitin pathway[21,27,46].

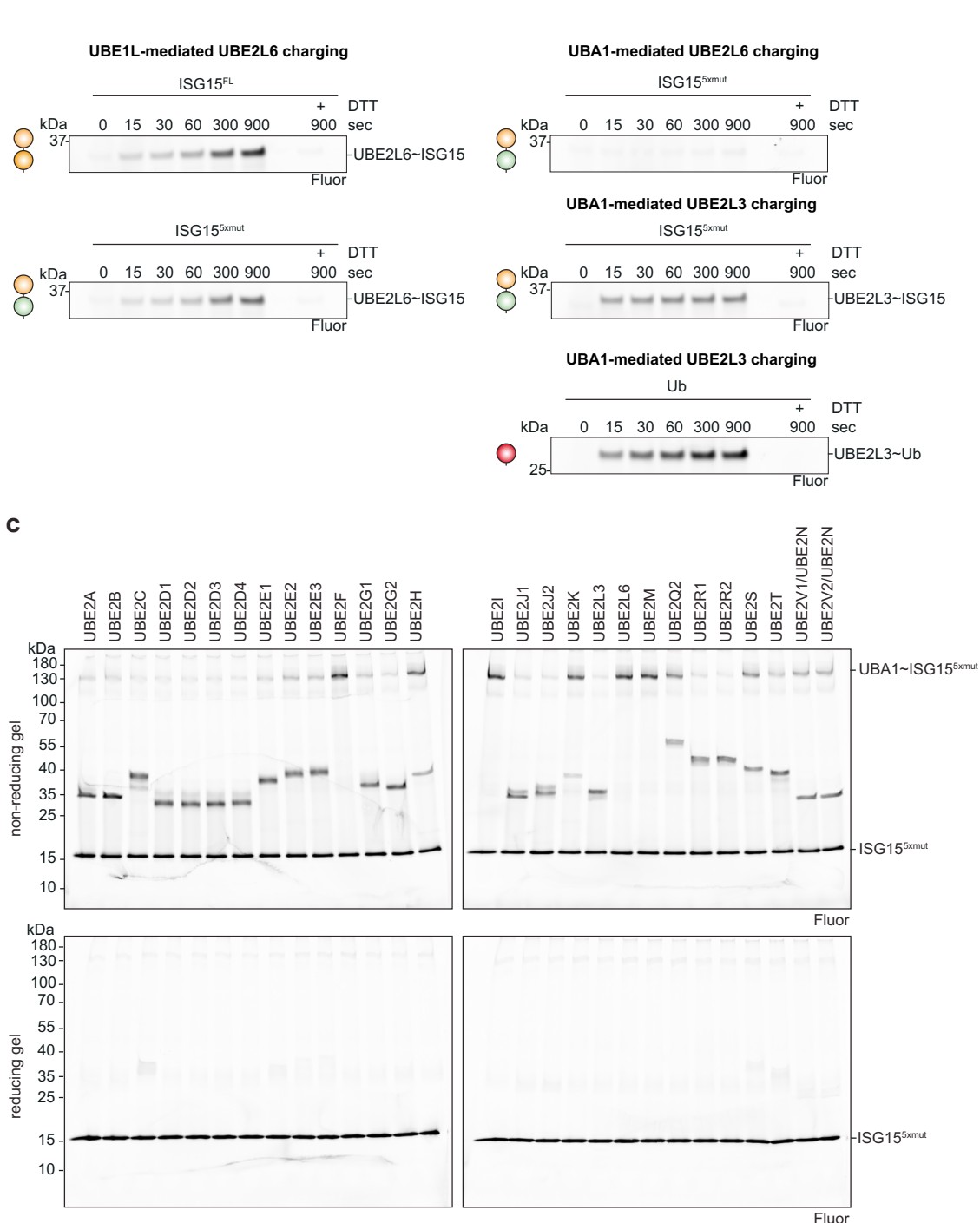

**Fig. 5 | Accessing ubiquitin E2 enzymes with ubiquitylized ISG15. a** Comparison of UBE1L-mediated UBE2L6 charging with ISG15 and ISG15^5xmut (W123R/T125I/P130Q/F149H/N151V). Reactions were quenched at the indicated time points, separated by SDS-PAGE, and visualized using fluorescent imaging. **b** Comparison of UBA1-mediated charging of UBE2L6 and UBE2L3 with ISG15^5xmut, and charging of UBE2L3 with ubiquitin (Ub). Reactions were visualized as in **a**. **c** Comprehensive analysis of UBA1-mediated E2 charging with ISG15^5xmut. All ubiquitin E2 enzymes tested formed a thioester bond with ISG15^5xmut, while E2 charging was not observed for SUMO (UBE2I), NEDD8 (UBE2F, UBE2M), and ISG15 (UBE2L6) E2 enzymes (also see Supplementary Fig. 8). Experiments were performed independently in triplicate. Source data are provided in the Source Data file.

Interestingly, crystal structures of the dual-specificity E1 enzyme UBA6 in complex with either ubiquitin, or with FAT10 were recently described[25,26]. Like ISG15, FAT10 contains two tandem ubiquitin-like domains fused through a short linker. Interestingly, the UBA6-FAT10 structure revealed that the FAT10 N-lobe contacts a defined interface of UBA6 (Supplementary Fig. 5e), and these interactions were

functionally relevant. While our data suggests the ISG15 N-lobe is not required for UBE1L activity in vitro, it will be important to determine the function(s) of this enigmatic domain. It seems likely that the N-lobe will be important for downstream components of the pathway, including substrate targeting[45]. Another intriguing possibility is that similar to linkage-specific ubiquitin-binding domains both ubiquitin-

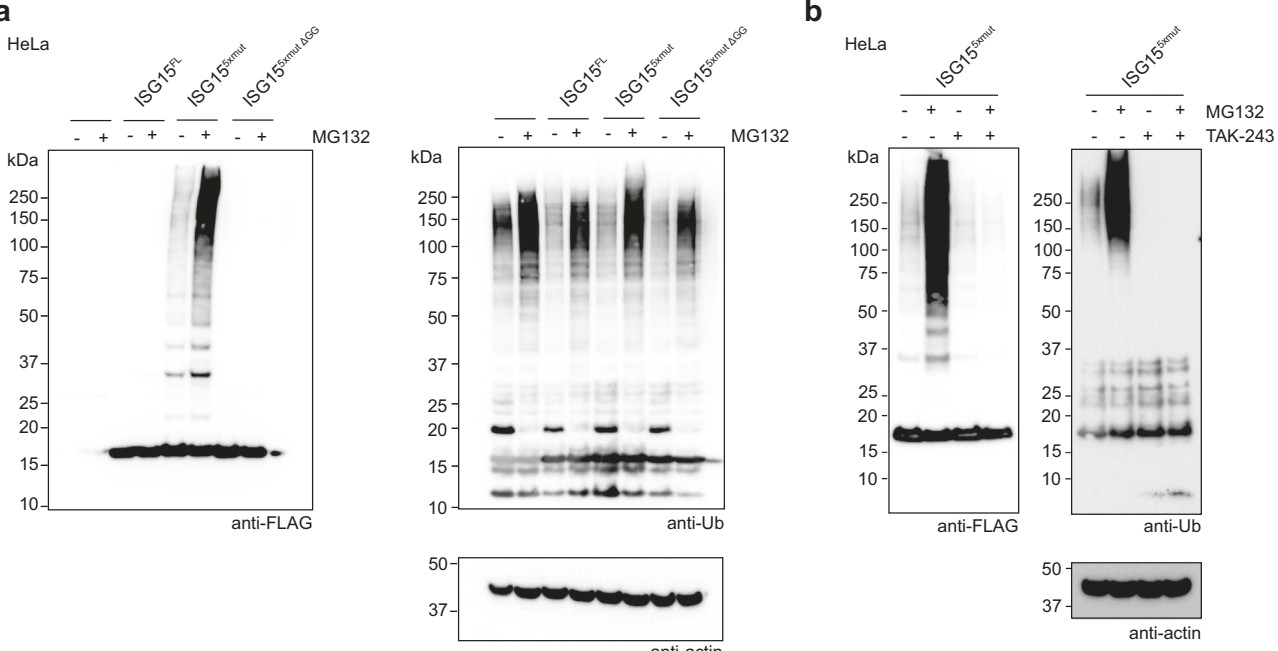

**Fig. 6 | Accessing the ubiquitin system with ubiquitylised ISG15 in cells.**
**a** Transfection assays with FLAG-tagged full-length ISG15 (ISG15[FL]), ISG15[5xmut] (W123R/T125I/P130Q/F149H/N151V), and a non-conjugatable version of ISG15[5xmut] (ISG15[5xmut] ΔGG). Cells were treated without or with the proteasomal inhibitor MG132. ISG15 substrate modification was monitored by anti-FLAG western blots. Anti-ubiquitin (Ub) and anti-actin western blots were performed as controls.

**b** Treatment of ISG15[5xmut] transfected cells with a ubiquitin E1 inhibitor. Cells were treated with DMSO, MG132, the UBA1 inhibitor TAK-243, or a combination of both MG132 and TAK-243. ISG15 substrate modification was monitored by anti-FLAG western blots. Anti-Ub and anti-actin western blots were performed as controls. Experiments were performed independently in triplicate. Source data are provided in the Source Data file.

like folds are required for recognition by ISG15 binding domains[47,48]. Moving forward, it will be important to combine the insights gained from structural studies of assembly and disassembly machineries, to help guide the identification of bona fide ISG15 binding domains.

In addition to insights into ISG15 recognition, we have also characterized several determinants of E2 selectivity into the ISG15 pathway. Similar to the ubiquitin and Ubl systems, E1-E2 interactions dictate E2 selection, rather than E2-ubiquitin/Ubl interactions[28,37,38,49–54]. Our structural and biochemical analysis characterized the hydrophobic interface between the UFD and α1 helix of UBE2L6 in molecular detail and described UBE2L6 mutants with enhanced activity with the ubiquitin machinery. Strategies to either enhance or limit this interaction could prove useful for studying the roles of ISG15 modifications in the antiviral response. However, the UBE2L6 mutants from our study remain to be tested in a cellular context. Nevertheless, we are excited by the possibility of using such an approach.

The visualization of the downstream steps of the ISG15 pathway remain to be elucidated; this includes the mechanisms of ISG15 E2-E3 transfer, substrate targeting, functional effects of substrate ISGylation, and recognition of modified substrates by ISG15 binding domains. Furthermore, it will be important to understand how bacterial and viral effector proteins interfere with the individual steps of the ISG15 enzyme cascade and the implications of these interactions on pathogenesis. Certainly, the adaptation of chemical biology approaches developed for ubiquitin research will be instrumental in capturing many of these intermediate reactions. However, given the remarkable complexity of the ubiquitin system as well as the emerging implications of ISG15 and ISGylation in numerous diseases, additional tools and approaches are likely needed to fully understand the details of the ISG15 system. The information gained from such studies will help define the rules of ISG15 recognition and transfer, and ultimately help us uncover the cellular functions of this important post-translational modification of the antiviral response.

## Methods
### Microbiology
ISG15, Ub/Ubl-ISG15 fusions, UBE2L6, UBE2L3, and NS1B constructs were cloned into the pCoofy3 (ISG15, Ub/Ubl-ISG15, UBE2L6 C86-only, NS1B), pCoofy5 (ISG15 C78S, UBE2L6), and pGEX (UBE2L3) bacterial expression vectors (pCoofy3 – N-terminal 6xHis-GST-3C, pCoofy5 – N-terminal 6xHis-SUMO1, pGEX – N-terminal GST-TEV). The SARS-CoV2 PLpro plasmid was a kind gift from D. Shin and I. Dikic[31]. UBE1L (1-1012aa) and HERC5 (638−1024aa) were cloned into the pLIB vector as N-terminal GST fusion with a TEV cleavage site[55]. A list of primers used in this study is provided in Supplementary Table 2.

### Protein expression and purification
Bacterial expression vectors were transformed into *Escherichia coli* Rosetta2 (DE3) pLacI cells. Cultures were grown to an $OD_{600}$ 0.6–0.8 and induced with 0.2 mM IPTG for 16 h at 18 °C. After centrifugation, pellets were resuspended in lysis buffer (50 mM Tris pH 7.5, 150 mM NaCl, 2 mM β-mercaptoethanol) and frozen at −80 °C. Once thawed, one tablet of ETDA-free Complete Protease Inhibitor (Roche) was added to the cell suspension. Samples were then sonicated, centrifuged, and the cell lysate was removed. Proteins were purified from lysate using either nickel (Sigma-Aldrich) and/or glutathione resin (GE Healthcare). Proteins for structural studies were eluted from resin using lysis buffer supplemented with imidazole and/or glutathione. Affinity tags were cleaved overnight at 4 °C using TEV or 3 C proteases. After cleavage, proteins were further purified using ion exchange and size-exclusion chromatography. Proteins for biochemistry studies, including those for mutational analysis, were eluted by proteolytic cleavage off the affinity resin. Eluted proteins were then re-incubated with fresh affinity resin to remove unwanted protease contaminants. UBE1L and HERC5 were expressed in High-Five cells (Thermo Fisher, Cat. no. B85502) infected with baculovirus prepared using Sf9 cells (Thermo Fisher, Cat. no. 11496015). After centrifugation, pellets were

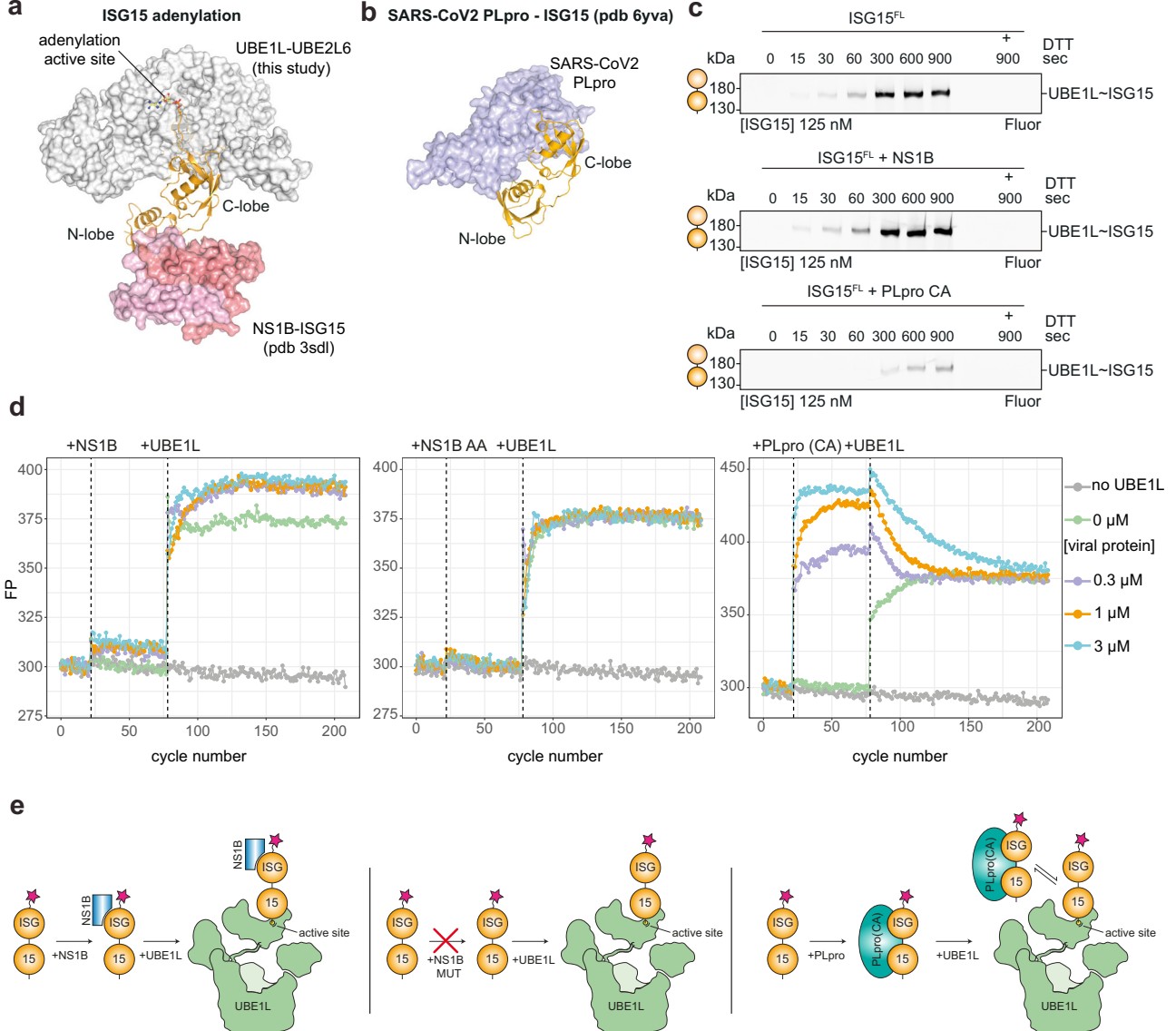

**Fig. 7 | The impact of viral proteins on ISG15 E1 charging. a** Structure of nonstructural protein 1 from influenza B virus (NS1B) bound to ISG15 (pdb 3sdl)[43] overlaid onto the UBE1L-UBE2L6 ISG15 adenylate complex. NS1B contacts the N-terminal ubiquitin-like fold (N-lobe) of ISG15, but not the C-terminal ubiquitin-like fold (C-lobe). The ISG15 adenylate and UBE1L adenylation active site are highlighted. **b** Structure of SARS-CoV2 papain-like protease (PLpro) bound to ISG15 (pdb 6yva)[31]. PLpro contacts both the N-lobe and C-lobe of ISG15. **c** Time course analysis of UBE1L charging assays in the presence of viral proteins. The catalytically inactive PLpro mutant Cys111Ala (CA) was used. **d** Analysis of UBE1L charging in the presence of viral proteins using fluorescence polarization (FP) assays. The indicated concentrations of NS1B, NS1B AA mutant (W36A/Q37A) and PLpro CA were added to fluorescent ISG15, followed by the addition of UBE1L to the sample. **e** Schematic of the proposed mechanisms to explain data shown in **c** and **d**. NS1B binds the N-lobe of ISG15 and remains bound during UBE1L charging, while PLpro CA competes with UBE1L for the C-lobe of ISG15. Experiments were performed independently in triplicate. Source data are provided in the Source Data file.

resuspended in lysis buffer. High-Five cells were resuspended in lysis buffer containing: 50 mM Tris pH 8.0, 150 mM NaCl, 5 mM DTT, 10 μg/mL leupeptin, 20 μg/mL aprotinin, 2.5 mM PMSF, 1 tablet of EDTA-free Complete Protease Inhibitor. Samples were then sonicated, centrifuged, and the cell lysate was removed. Proteins were purified using glutathione resin (GE Healthcare) and eluted with lysis buffer supplemented with glutathione. Affinity tags were removed by proteolytic cleavage overnight at 4 °C using the TEV protease. After cleavage, proteins were further purified using ion exchange and size-exclusion chromatography.

### Formation of UBE1L ISG15(A) complex

The UBE1L adenylated ISG15 (UBE1L-ISG15(A)) complex was assembled by incubating 60 μM of catalytically inactive UBE1L (Cys599Ala) with 240 μM of ISG15 in binding buffer (50 mM Tris pH 8.0, 150 mM NaCl, 2 mM DTT, 10 mM MgCl₂, 10 mM ATP) for 30 min at 25 °C. The sample was then loaded onto an ÄKTAmicro system (GE Healthcare) and separated using a Superdex 200 Increase 3.2/300 column (GE Healthcare), which was pre-equilibrated in binding buffer. Fractions were collected and analyzed by Coomassie-stained SDS-PAGE gels and fractions containing both UBE1L (C599A) and ISG15 were pooled and concentrated to ~2 mg/ml. The concentrated sample was used for cryo-EM analysis.

### Formation of UBE1L-UBE2L6 ISG15(A) complex

E1-E2 cross-linking was performed as previously described with minor modifications[18]. To eliminate non-specific disulfide bonds, surface exposed UBE2L6 cysteine residues (Cys98 and Cys102) were mutated

to serine (as described in Olsen and Lima, 2013). This mutant, referred to as UBE2L6[C86only], allowed for site-specific disulfide bond formation between the E1 and E2 active sites. To 'activate' E2 enzymes before cross-linking, E2s were buffer exchanged into cross-linking buffer (20 mM Tris pH 8.0, 50 mM NaCl), then mixed at a 1:1 (vol/vol) ratio with activation buffer (20 mM Tris pH 8.0, 50 mM NaCl, 2.5 mM 2,2'-Dipyridyl disulfide, 2.5% DMSO), and incubated for 30 min at 25 °C. E2 proteins were then buffer exchanged back into cross-linking buffer for use in cross-linking reactions. UBE1L was also buffer exchanged into cross-linking buffer. In small-scale cross-linking reactions (for example, Supplementary Fig. 2a), 1 μM UBE1L was incubated with 2.5 μM 'activated' UBE2L6 for 15 min at 25 °C. In cross-linking experiments for cryo-EM analysis, 5 μM UBE1L was incubated with 12.5 μM 'activated' UBE2L6 for 15 min at 25 °C. To form the complex, UBE1L-UBE2L6 was incubated with 10 μM ISG15 Cys78Ser, 5 mM MgCl₂, and 2.5 mM ATP prior to separation by size-exclusion chromatography using a Vanquish HPLC (Thermo Fisher). Fractions were collected and analyzed by Coomassie-stained SDS-PAGE gels. Fractions containing the UBE1L-UBE2L6 cross-linked complex and ISG15 were pooled and concentrated to ~1 mg/ml. The concentrated sample was used for cryo-EM analysis.

## Cryo-EM

**Sample preparation.** Prior to sample application, R1.2/1.3 holey carbon grids (Quantifoil) were cleaned using a plasma cleaner for 30 s on medium strength. For the UBE1L-ISG15(A) complex, the sample was diluted to 1 mg/ml and 3 μL was applied to the grid. For the UBE1L-UBE2L6 ISG15(A) complex, the sample was diluted to 0.25 mg/ml and 3 μL was applied to the grid. After sample application, grids were immediately plunged into liquid ethane using a Vitrobot Mark IV (Thermo Fisher). Vitrobot settings were as follows: humidity, 100%; blotting time, 3 s; blotting force, 5.

**Electron microscopy.** Screening of grids and initial data collection was performed on a Glacios cryo transmission electron microscope equipped with a K2 Summit direct detector. For the UBE1L ISG15(A) complex, 2088 images were recorded at 1.479 Å per pixel with a nominal magnification of 28,000x. A total dose of 60 e⁻ Å⁻² was used over 37 frames, with a defocus range from −1.2 to −3.0 μm. For the UBE1L-UBE2L6 ISG15(A) complex, 2136 images were recorded at 1.885 Å per pixel with a nominal magnification of 22,000x. A total dose of 60 e⁻ Å⁻² was used over 40 frames, with a defocus range from −1.2 to −3.0 μm.

After analysis of the UBE1L-UBE2L6 ISG15(A) complex on the Glacios, a duplicate grid was collected using the high-resolution Titan Krios electron microscope at 300 kV equipped with a post-column gatan imaging filter and a K3 Summit direct detector in counting mode. In total, 12,326 images were recorded at 0.8512 Å per pixel with a nominal magnification of 105,000x. A total dose of 61.5 e⁻ Å⁻² was used over 40 frames, with a defocus range from −1.2 to −3.6 μm.

**Data processing.** Images were processed using Relion3.0/3.1[56]. Drift correction and defocus estimation were performed using MotionCorr2[57] and CTFFIND v4.1.9[58] or Gctf-1.06 (Kai Zhang, MRC Laboratory of Molecular Biology). For the UBE1L ISG15(A) dataset, particles were picked with Gautomatch v0.56 (Kai Zhang, MRC Laboratory of Molecular Biology). For the UBE1L-UBE2L6 ISG15(A) dataset, a template for Gautomatch picking was generated using EMAN2[59] from a 3D model generated from the Glacios dataset. All subsequent data processing steps were performed in Relion (e.g. 2D classification, 3D classification, 3D Refinement; see Supplementary Figs. 3 and 4 for processing workflow). Post-processing was performed using both Relion (Fig. 2a–c, Fig. 3) and DeepEMhancer (Fig. 2d)[60].

**Model building.** The initial model for the UBE1L-UBE2L6 ISG15(A) complex was generated using Chimera v1.16[61]. Briefly, a UBE1L

structural model was predicted with a high-degree of confidence using Phyre2.0[62], while ISG15 (pdb 6FFA)[34] and UBE2L6 (pdb 1WZV; were downloaded from the Protein Data Bank[34,63]. Using Chimera, structures were manually placed into the cryo-EM density and adjusted using rigid-body refinement. This initial model was subsequently used as a template for manual modeling building in Coot, and iterative refinements were performed in Coot v0.89 and Phenix v1.15.2[63]. Further structural analysis was performed using PyMOL v2.5.2 and ChimeraX v1.3.

## ISG15 fluorescent labeling

ISG15 and ISG15 mutants were cloned into pCoofy3 with a N-terminal cysteine immediately following the 3 C cleavage site. To specifically label the N-terminus, the only cysteine in ISG15 (Cys78) was mutated to a serine residue. Prior to labeling, purified proteins were diluted to 200 μM in dilution buffer (50 mM Tris pH 7.5, 150 mM NaCl, 10 mM DTT) and incubated for 10 min at 25 °C. Proteins were then buffer exchanged into 50 mM HEPES pH 7.5, 150 mM NaCl using Zeba columns (2x, Thermo Fisher). To start the labeling reaction, a 10-fold molar excess of BODIPY fluorescent maleimide (Thermo Fisher) was added to the proteins and incubated for 2 h at 25 °C in the dark. After 2 h, the labeling reaction was stopped with the addition of 10 mM DTT. Proteins were buffer exchanged into 50 mM Tris pH 7.5, 150 mM NaCl, 2 mM DTT using Zeba columns (2x, Thermo Fisher) and stored at −80 °C until later use in biochemical assays.

## E1 charging assays

For assays in Fig. 4c, e–g, Fig. 7c, and Supplementary Fig. 7b, c, e, f, the E1 enzyme was used at a concentration of 2 μM and the ISG15 concentrations are indicated in the figures. In Fig. 4d and Supplementary Fig. 7d, E1 enzyme was used at a concentration of 1 μM. In Fig. 7c, viral proteins were used at a concentration of 4 μM. Reactions were performed in charging buffer (50 mM HEPES pH 7.5, 150 mM NaCl, 5 mM MgCl₂, 5 mM ATP). For the ATP-dependency assays (Supplementary Fig. 7b), the concentration of ATP varied (0 to 5 mM ATP, as indicated in the figure) and the reactions were stopped after 5 min. Reactions were performed in buffer containing 50 mM Tris pH 7.5, 150 mM NaCl, 10 mM MgCl₂. All reactions were performed at 25 °C, quenched with LDS sample buffer at the indicated time points, separated by SDS-PAGE, and visualized using fluorescent imaging (Amersham Typhoon) and/or Coomassie staining (Amersham Imager 600). In Fig. 4e, f, and Supplementary Fig. 7e, quantification of E1 charging was performed using ImageQuant v5.2 and plotted in GraphPad Prism v9.3.1. Uncropped and unprocessed gel images are provided in the Source Data file.

## Viral protein binding assays

Fluorescence polarization binding assays were performed in a black round-bottom 384-well plate using fluorescent ISG15 and WT UBE1L. ISG15 was prepared at a 1x concentration of 100 nM in assay buffer (50 mM HEPES pH 7.5, 150 mM NaCl, 10 mM MgCl₂, 5 mM ATP). Viral proteins (NS1B, NS1B 36 A/37 A, PLpro CA) and UBE1L (20 μM) were prepared at a 20x concentration in assay buffer. 1 μL of each protein was added to the fluorescent ISG15 mixture at indicated cycle number. Fluorescence polarization measurements were made with a 482 nm excitation filter and a 530 nm emission filter at a target gain of 300 using a PHERAstar Plus plate reader (BMG Labtech). 30 flashes per well were performed every 15 seconds. Data was analyzed using R studio (version 4.2.0).

## E2 charging assays

Single-turnover E2 charging assays were performed as previously described with minor adjustments[27]. For pulse reactions, UBE1L - ISG15 complexes were formed by incubating 2.5 μM UBE1L with 3 μM fluorescent ISG15 in charging buffer (50 mM HEPES pH 7.5, 150 mM NaCl,

5 mM MgCl$_2$, 5 mM ATP) for 15 min at 25 °C. For UBA1-ISG15$^{5xmut}$ pulse reactions, the incubation time was 15 min in Fig. 5b and 60 min in Fig. 5c at 25 °C. Reactions were stopped by diluting the sample 10-fold in quenching buffer (50 mM HEPES pH 7.5, 150 mM NaCl, 100 mM EDTA). To initiate ISG15 transfer from UBE1L to the E2 enzyme (chase reaction), 2 μM E2 enzyme was added to the preformed E1 thioester complexes. Samples were collected at the indicated time points and mixed with 3x LDS sample buffer. Proteins were separated by SDS-PAGE gels and visualized using fluorescent imaging. For multi-turnover assays in Supplementary Fig. 2b and Supplementary Fig. 10a, ISG15 (5 μM) was mixed with UBE1L (0.25 μM) and UBE2L6 (2 μM) in buffer containing 50 mM Tris pH 7.5, 150 mM NaCl, 10 mM MgCl$_2$, 10 mM ATP. Reactions were performed at 25 °C, quenched with 3x LDS sample buffer at the indicated time points, separated by SDS-PAGE, and visualized using Coomassie stain (Amersham Imager 600) and/or fluorescent imaging (Amersham Typhoon). Uncropped and unprocessed gel images are provided in the Source Data file.

### In vitro ubiquitin assembly reactions

For ISG15 assembly reactions in Supplementary Fig. 2c, ISG15 (25 μM) was mixed with UBE1L (1 μM), UBE2L6 (10 μM), and HERC5 638–1024aa (10 μM) in buffer containing 50 mM Tris pH 8.0, 150 mM NaCl, 5 mM MgCl$_2$, 5 mM ATP, 0.5 mM DTT. For ubiquitin assembly reactions in Supplementary Fig. 10f, ubiquitin (10 μM) was mixed with UBA1 (0.5 μM), E2 enzymes (2.5 μM), and GST-tagged NEDD4L 576–995aa (2.5 μM) in buffer containing 50 mM HEPES pH 7.5, 150 mM NaCl, 5 mM MgCl$_2$, 5 mM ATP, 0.5 mM DTT. Reactions were performed at 28 °C for ISG15 assembly reactions and 37 °C for ubiquitin assembly reactions, quenched with LDS sample buffer at specific time points, separated by SDS-PAGE, and visualized using Coomassie stain (Amersham Imager 600), fluorescent imaging (Amersham Typhoon), and anti-ubiquitin Western blots (Bio-Rad ChemiDoc). Uncropped and unprocessed gel images and blot are provided in the Source Data file.

### Tissue culture, transfections, and western blotting

HeLa cells (CCL-2) obtained from ATCC were maintained in DMEM media supplemented with 2 mM L-Glutamine, 100 U/ml Pen/Strep, and 10% FBS at 37 °C in 5% CO2. Cells were grown to 80% confluency prior to transfection with the indicated pcDNA5 plasmids (FLAG tagged ISG15, HA tagged ISG15 enzymes (UBE1L, UBE2L6$^{WT}$ or UBE2L6$^{C86only}$, HERC5$^{(31-1024)}$) using the FuGene transfection reagent (Promega). For MG132 treatment, after 24 h, cells were treated with either DMSO or 10 μM MG132 for 4 h. For TAK-243 treatment, after 24 h, cells were treated with either DMSO or 500 nM TAK-243 for 5 h. For co-treatment with both MG132 and TAK-243, after 24 h, cells were first treated with 500 nM TAK-243 for 1 h followed by the further addition of DMSO or 10 μM MG132 for 4 h. Cells were lysed by adding 2x LDS sample buffer containing 10 mM DTT, followed by sonication to shear DNA. Samples were resolved by SDS-PAGE followed by transfer onto nitrocellulose membranes using the Trans-Blot Turbo Western blotting system (Bio-Rad). Blots were probed with FLAG (Merck, F3165), ubiquitin (Novus, NB300-130SS), UBE1L (SantaCruz, sc-390097), HA (abcam, ab18181), HERC5 (Invitrogen, PA5-100555) and actin (ProteinTech, 66009) primary antibodies at a concentration of 1:1000 or 1:5000 (actin) in 5% fat free milk in TBS with 0.1% Tween. Following incubation with primary antibodies, blots were incubated with either mouse- or rabbit-HRP conjugated secondary antibodies (Thermofisher 31450 and 31460, respectively) at a concentration of 1:5000. Blots were visualized using SuperSignal chemiluminescence substrate (ThermoScientific) and a ChemiDoc MP imaging system (Bio-Rad). Uncropped and unprocessed blots are provided in the Source Data file.

### Reporting summary

Further information on research design is available in the Nature Portfolio Reporting Summary linked to this article.

## Data availability

Cryo-EM maps have been deposited to the Electron Microscopy Data Bank (EMDB) as UBE1L - UBE2L6(C98S/C102S) ISG15(C78S) accession numbers EMDB-16891 and EMDB-18589 and atomic coordinates are available through to the Research Collaboratory of Structural Bioinformatics Protein Data Bank (RCSB PDB) as accession code 8OIF. Published PDB accession codes used for analysis are available under the following accession codes: 6FFA, 1Z2M, 1WZV, 3SDL, 6YVA, 4NNJ, 1R4N, 1Y8R, 7PYV, 6DJX, 4II2, 6NYA. Raw uncropped images for all SDS-PAGE gels and western blots are in the Source Data section. Source data are provided with this paper. The authors declare there are no restrictions on the data availability of the research presented within this study and all material will be available upon request from the corresponding authors. Source data are provided with this paper.

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

## Acknowledgements

We thank the members of the Schulman and Virdee Labs for reagents and helpful discussions and D. Bollschweiler, T. Schäfer, and the cryo-EM facility at the Max Planck Institute of Biochemistry. This study was supported by the Max Planck Gesellschaft, EU H2020 ERC Advanced Grant (Nedd8Activate, Grant agreement ID: 789016), SFB 1035 (German Research Foundation DFG, Sonderforschungsbereich 1035, Projektnummer 201302640, project A13) to B.A.S and the Medical Research Council (MC_UU_00018/10) and Royal Society (RGS_R2_222185) to K.N.S. K.N.S. is a Lister Institute Prize Fellow.

## Author contributions

K.N.S. conceived and designed the study, performed experiments, interpreted results, wrote the manuscript, and acquired funding. I.W. designed assays, performed experiments, interpreted results, and edited the manuscript. K.B. and J.R.P. collected, refined, and processed cryo-EM data. R.V. and S.V.G prepared reagents. B.A.S. conceived and designed the study, interpreted results, wrote the manuscript, and acquired funding.

## Competing interests

B.A.S. is a member of the scientific advisory boards of Interline Therapeutics and BioTheryX. The remaining authors declare no competing interests.
