## [Peer Review File · Nature Communications]

REVIEWER COMMENTS

Reviewer #1 (Remarks to the Author):

Wallace and colleagues describe the first steps of the post-translational modification ISGylation. The authors use a structural biology technique to pinpoint the determinants of the E1 and E2 enzymes involved in ISGylation towards ISG15, to really define what makes them specific for ISG15 instead of Ub. This is an important addition to the literature so that we can start understanding the how this posttranslational modification is initiated. Many aspects of this process remain unclear, including the transfer from E2 to E3, the specificity of USP18 as a deISGylase (and how it differs from the interaction to be stabilized by ISG15), and the specificity of ISG15 for different substrates (which also include non-viral proteins). The paper by Wallace et al describes the first step of the process in part.

- Authors should be more careful in their literature assertions. Actually, under native conditions (not over expression systems) there is very little evidence that ISGylation is actually antiviral, and more often than not no effect on viral replication in vitro, and often in vivo in animals is noted. In humans there is a stark difference in how ISG15 functions. Thus stating "The attachment of the ubiquitin-like protein ISG15 to substrates is a well-established antiviral signalling mechanism of the innate immune response" has a few caveats. ISGylation system can be forced to be antiviral. Author should acknowledge this, species differences and expand on the phenotypes in individuals with loss of ISG15 (and thereby ISGylation), and patients with loss of USP18 (and thereby having hyper ISGylation, and defective deISGylation, or hypomorphic USP18 patients with normal deISGylation), as those can also inform potential roles for ISGylation. I would suggest more balanced approach to literature and maybe focus on structural determinants more.

- The main concern of this reviewer is the overreliance on the artificial tools.

- The need for E2 to stabilize the complex to be able to resolve the structure of ISG15 interacting with UBE1L. As a curiosity, how do other authors show interaction of Ub with E1? What is different between these systems?

- The need for mutated cysteine residues. Are these mutants equally functionally active to their WT counterparts? If transfected together with ISG15, UBE1L and HERC5, would the authors be able to detect the same ISGylated proteins?

- What are the specific determinants of UBE1L-UBE2L6 vs other E1-E2? Supplemental figure 6e shows interaction of UBE1L-UBE2L6 vs UBE1L-UBE2L3. The authors claim that the later forms a less productive interaction interface. How is this interaction compared to an E1 specific for UBE2L3?

- In figure 5 the authors very well describe that a very “ubiquitylized” ISG15, is able to ISGylate using the ubiquitination machinery. Is this ISG155xmut able to ISGylate using the canonical ISGylation machinery? The authors should transfect ISG15 (FL vs 5xmut vs dGG) together with UBE1L, UBE2L6 and HERC5 in the presence of a ubiquitination inhibitor.
- Most gels need to be run together to be able to compare the results. For example, in figure 5a and b, unless run together, it is not possible to really claim that the rates of UBE2L3 and UBE2L6 are comparable. In this figure in particular, as presented, it actually looks like UBA1-mediated UBE2L3 charging with ISG155xmut is enhanced compared to the other gels. Other experiments should also be run together: Supp fig 9d, supp fig 7f and g, panel in figure 4.
- In figure 5 the authors should also compare the UBA1-mediated UBE2L3 in when interacting with Ub.
- Some experiments lack positive or negative controls, figure 4h for example.
- Supplemental figure 8a left panel should include UBE2L6 as a control

Reviewer #2 (Remarks to the Author):

The manuscript by Wallace et al. reported the cryo-EM structures of UBE1L in complex with UBE2L6 and activated ISG15 which provides insights into the initial catalytic steps of the ISG15 conjugation pathway. They trapped a stable, yet functional complex of cross-linked UBE1L-UBE2L6 along with adenylated ISG15. Structural-based mutagenesis and biochemical experiments are appropriately performed and are supportive of the importance of the catalytically relevant residues and confirms the molecular basis of the UBE1L specificity for ISG15 and UBE2L6. I think this manuscript should be published, I only have minor comments which I listed here.

Comments:

1. While testing the effect of viral effectors, authors have used Influenza WT NS1B and SARSCoV2 PLPpro activity dead mutant in the UBE1L self-conjugation reaction. I do not really understand the reason behind testing the PLPpro activity dead mutant in this reaction since they began testing the viral effectors which should be WT. Does the WT PLPpro inhibit UBE1L activation?
2. Authors nicely show how ISG15 patch mutants can be made UBA1 compatible. On a similar note, from the paper it does seem like a natural step to prepare Ube2L6 specific mutations to make it more like Ube2L3 and test if it then becomes compatible with UBA1 or there are other specificity determinants.

Minor comments:

1. In the line 200-201, it is stated "Another potential interesting contact site surrounds of UBE2L6, which based on cryo-EM density contacts the SCCH domain" but a figure for this is missing?
2. Line no. 108- authors should mention that they used C599A mutant of UBE1L
3. Line no. (175-177): It would be informative to have a structure figure with overlapping AMP orientation and identical active site residues
4. Line 188-189: "In particular, ISG15's Trp123 and P130 create a unique hydrophobic patch adjacent to Ub's Ile44 patch and forms additional interactions with the hydrophobic beta-sheet." It would be better if the authors also mention the interacting residues in UBE1L.
5. Line 190-191: "Importantly, these hydrophobic contacts do not exist in the UBA1-ubiquitin complex but are instead replaced with ionic interactions." Which residues are involved in these ionic interactions? In the suppl fig. 5f, we can only see the residues on Ub (R42 and Q49), but not on the UBA1.
6. Line No.- 195-196: "Modelling of a phylogenetically related ubiquitin E2, UBE2L3, onto UBE2L6 helped provide insight into this specificity" It would be informative to mention the sequence similarity.
7. The y-axis is not labeled in suppl fig 3; fsc curve
8. Supp Fig 9c- the catalytic Cys is Cys86 (seq alignment figure 9b), but in the structure figure 9c it is labeled as C85.

REVIEWER COMMENTS

We would like to thank both reviewers for their time and constructive feedback on our manuscript. We have taken on board all points from both reviewers and feel that our resubmitted manuscript has been made stronger by the inclusion of the suggested experiments. While we were addressing the comments below to further improve our manuscript, and after our submission to Nature Communications and pre-print upload to bioRxiv, a competing manuscript was submitted and published which also elucidated a structure for UBE1L (UBA7) in complex with UBE2L6 and ISG15 (Afsar et. al, Nature Communcations), which we have now acknowledged in our revised manuscript. Their work reveals similar conclusions to ours, which we feel increases the robustness of our work. Furthermore, we feel our work goes well beyond theirs in terms of defining the molecular basis of understanding Ub/ISG15 specificity by using biochemical and cellular studies, which we have now further expanded upon as suggested by the reviewers in their comments below. Throughout the revision process we have remained determined to maintain and improve upon the quality of our work by taking on board the reviewer's constructive feedback, performing all experiments requested to a high standard, and improving the manuscript as suggested. We again would like to thank the reviewers for their time and consideration.

Reviewer #1 (Remarks to the Author):

Wallace and colleagues describe the first steps of the post-translational modification ISGylation. The authors use a structural biology technique to pinpoint the determinants of the E1 and E2 enzymes involved in ISGylation towards ISG15, to really define what makes them specific for ISG15 instead of Ub. This is an important addition to the literature so that we can start understanding the how this posttranslational modification is initiated. Many aspects of this process remain unclear, including the transfer from E2 to E3, the specificity of USP18 as a deISGylase (and how it differs from the interaction to be stabilized by ISG15), and the specificity of ISG15 for different substrates (which also include non-viral proteins). The paper by Wallace et al describes the first step of the process in part.

- Authors should be more careful is their literature assertions. Actually, under native conditions (not over expression systems) there is very little evidence that ISGylation is actually antiviral, and more often than not no effect on viral replication in vitro, and often in vivo in animals is noted. In humans there is a stark difference in how ISG15 functions. Thus stating "The attachment of the ubiquitin-like protein ISG15 to substrates is a well-established antiviral signalling mechanism of the innate immune response" has a few caveats. ISGylation system can be forced to be antiviral. Author should acknowledge this, species differences and expand on the phenotypes in individuals with loss of ISG15 (and thereby ISGylation), and patients with loss of USP18 (and thereby having hyper ISGylation, and defective deISGylation, or hypomorphic USP18 patients with normal deISGYlation), as those can also inform potential roles for ISGylation. I would suggest more balanced approach to literature and maybe focus on structural determinants more.

We would like to thank reviewer #1 for the positive assessment of our work and for highlighting the importance of this study to the field. We are also grateful for this reviewer's helpful suggestions on how we can further improve the text to cover the literature more

comprehensively. ISG15's species-specific differences and the identification of individuals with ISG15 and USP18 deficiencies are landmark discoveries, and we agree that these points should be discussed. Therefore, in the resubmitted manuscript, we have updated the text to emphasise the critical functional differences between human and mouse ISG15 signalling. In addition, we have discussed how ISG15 and USP18 deficient patients can help us understand the molecular and cellular functions of ISG15. By making these improvements and by addressing the comments below, we hope reviewer#1 will continue to be supportive of our manuscript.

(1) The main concern of this reviewer is the overreliance on the artificial tools.

We understand this concern. However, the tools and approaches used in this study were validated before performing these experiments. The reagents that fall into this category include the UBE2L6 Cys86-only mutant used to capture the E1-E2 complex, fluorescently labelled ISG15 for gel-based and fluorescence polarisation assays, and FLAG-tagged ISG15 for transfection assays. Below we discuss how we validated these tools.

(A) UBE2L6 Cys86-only mutant - To validate this reagent for structural studies we compared the enzymatic activity of the UBE2L6 Cys86-only mutant to the wild-type UBE2L6 enzyme. Using ISG15 E2 charging as a readout of activity, we show that the UBE2L6 mutant was active and displayed slightly reduced rates of E2 charging (Supplementary Fig. 2b). Furthermore, we confirmed that the UBE2L6 Cys86-only mutant labels the active site cysteine of UBE1L, while the wild-type enzyme produced a smear of unwanted disulfide bonds, which due to the heterogeneity of the sample would have complicated, if not prevented, structural analysis. Finally, we have also assessed the activity of this UBE2L6 mutant in cells (see point 3 below).

(B) Fluorescent ISG15 – To determine if fluorescent ISG15 interferes with the ISG15 conjugation pathway, we performed several controls. First, we compared ISGylation reactions with unlabelled and labelled ISG15 and observed no major differences in activity (see below, and Supplementary Fig. 2c in the updated version of the manuscript). Secondly, for fluorescence polarisation (FP) assays, we tested a NS1B mutant with defective ISG15 binding (i.e., NS1B AA; see Fig. 7d) and E1/E2 charging reactions without the addition of ATP (unpublished data), and observed no change in FP signal, thereby demonstrating that the BODIPY fluorophore is not resulting in non-specific protein-protein interactions. Together, these controls confirm the suitability of fluorescent ISG15 for studying the ISG15 conjugation pathway.

(C) Flag-tagged ISG15 – In order to validate our biochemical data which showed that the ISG15 5xmutant is transferred through ubiquitin E1/E2 enzymes we used Flag-tagged ISG15 to monitor substrate modifications in cells. Here we transfected three different Flag-tagged ISG15 constructs (i.e., wild-type, 5xmutant, and 5xmutant lacking the C-terminal glycine residues $-\Delta GG$). The 5xmutant confirmed our biochemical data, while wild-type ISG15 and the ISG15 5xmutant ΔGG served as controls. Wild-type ISG15 controlled for any potential over-expression issues and the ISG15 5xmutant ΔGG helped rule out the possibility that the 5xmutant itself was being ubiquitinated. Together, these results confirm the validating of this reagent and approach for studying the 5xmutant's altered specificity.

(2) The need for E2 to stabilize the complex to be able to resolve the structure of ISG15 interacting with UBE1L. As a curiosity, how do other authors show interaction of Ub with E1? What is different between these systems?

There are several ways to stabilise and solve structures of Ub/Ubl E1 complexes. Here we attempted two. The first involves mutation of UBE1L's active site cysteine to alanine, thus capturing the initial step of ISG15 activation/adenylation. This strategy has been successfully used to determine several Ub/Ubl E1 structures, but using X-ray crystallography whereby the crystal lattice may provide stability to flexible regions (for example for the NEDD8 E1, Walden et al., 2003, Mol Cell, and SUMO1 E1, Lois and Lima, 2005, EMBO J). However, this strategy proved less useful for cryo-EM, presumably due to the intrinsic conformational heterogeneity of the complex. Therefore, we rationalised that stabilisation of the E1 domains – namely the SCCH and UFD domains – through E2 binding would help with cryo-EM analysis. Here we adapted a technique first published by Olsen and Lima, in which the Ub E1-E2 active site cysteines were cross-linked through a disulfide bond (Olsen and Lima, 2013, Mol Cell). The rationale behind this approach is that this complex would mimic how the active sites cysteines come together during the Ub/Ubl thioester transfer reaction. More recently, this approach was used to solve a structure between a Ub E1 and the E2 enzyme Cdc34 (Williams et al., 2019, Nature Comms). These two reports established this technique as a way of faithfully capturing E1-E2 Ubl interactions. Our study demonstrates this also works for the ISG15 system, and can be used for structural determination by cryo-EM. We also note that our conclusions regarding ISG15 E1-E2 interactions agree with those published while our work was under revision (Afsar et al., 2023, Nature Comms).

(3) The need for mutated cysteine residues. Are these mutants equally functionally active to their WT Counterparts? If transfected together with ISG15, UBE1L and HERC5, would the authors be able to detect the same ISGylated proteins?

To address this comment, we performed the experiment suggested by the reviewer. The results from this study demonstrate that the UBE2L6 C86-only mutant is active in cells and generates similar ISGylation levels compared to the wild-type enzyme (see panel to the right, and Supplementary Fig. 2d in the updated version of the manuscript). This result agrees with our biochemical data in Supplementary Fig. 2b, which shows the UBE2L6 mutant retains activity.

(4) What are the specific determinants of UBE1L-UBE2L6 vs other E1-E2? Supplemental figure 6e shows interaction of UBE1L-UBE2L6 vs UBE1L-UBE2L3. The authors claim that the later forms a less productive interaction interface. How is this interaction compared to an E1 specific for UBE2L3?

Regarding the first point, this is a great question, and one of the goals of this study. In Supplemental Fig. 6e, we do claim that based on our structure and a UBE1L-UBE2L3 model, UBE1L-UBE2L3 would form a less productive interface. We then go on to validate this claim by substituting these residues in UBE2L6 (Met5 & Val8) with the equivalent residues of UBE2L3 (Arg5 & Met8), which substantially decreases UBE1L-mediated charging as compared to the wild-type E2 (please see Supplementary Fig. 10d). Regarding the second question, this is more difficult to address since there is not a reported structure of UBE2L3 bound to a ubiquitin E1 enzyme and we feel to address this question structurally is beyond the scope this manuscript. Still, in the updated version of the manuscript, we have included a biochemical study that demonstrates that a UBE2L6 mutant containing mutations at this interface (UBE2L6^{M5R/V8M/M123A}) results in increased activity in assembly reactions with the ubiquitin machinery (see Supplementary Fig. 10f & point 2 from Reviewer #2 below). Furthermore, based on this comment, we have compared UBA1-mediated E2~Ub charging with UBE2L3, UBE2L6, UBE2L6^{M5R/V8M}, and UBE2L6^{M5R/V8M/M123A}. The UBE2L6 mutants show a slight increase in E2~Ub formation compared to wild-type UBE2L6, confirming these residues are involved in UBA1-UBE2L3 recognition (see panel to the right, or Supplementary Fig. 10e).

(5) In figure 5 the authors very well describe that a very “ubiquitylized” ISG15, is able to ISGylate using the ubiquitination machinery. Is this ISG15 5xmut able to ISGylate using the canonical ISGylation machinery? The authors should transflect ISG15 (WT vs 5xmut vs dGG) together with UBE1L, UBE2L6 and HERC5 in the presence of a ubiquitination inhibitor.

As suggested, we transfected ISG15 (wild-type, 5xmutant, 5xmutant Δ GG) and the ISG15 machinery into HeLa cells and treated these cells with either TAK-243, an E1 inhibitor with high specificity towards UBA1 (UBA1 IC50 - 1 nM vs. UBE1L IC50 - 5 μ M), or TAK-243 and MG132. We also performed the TAK-243/MG132 treatment in the absence of UBE1L, which enabled us to determine if the 5xmutant is passing through the ISG15 machinery. The results from this study show that the ISG15 5xmutant can ISGylate using the ISG15 enzyme cascade, since in the TAK-243/MG132 minus UBE1L treatment the levels of ISG15 modified substrates decreases, as compared to the same transfection assays plus UBE1L (see panel to the right, and Supplementary Fig. 9). These results are in agreement with our biochemical data, which shows that the ISG15 5xmut can form a thioester intermediate with UBE1L and UBE2L6 (see Fig. 4g, Fig. 5a, Supplementary Fig. 7e, f, and Supplementary Fig. 8).

Furthermore, to strengthen our finding that the ISG15 5xmutant is loading through the ubiquitin E1, we again used the Ub E1 inhibitor, TAK-243. In this experiment, HeLa cells were first transfected with FLAG-ISG15 5xmutant. TAK-243 was then added to cells, before the addition of MG132. As in our previous results, the FLAG-ISG15 5xmutant generates a high molecular weight smear of modified substrates – an effect which is significantly increased upon MG132 treatment. TAK-243 pre-treatment, however, prevented the accumulation of these modified substrates (see panel to the left, and Fig. 6b). Furthermore, and as expected, ubiquitination levels were drastically reduced following TAK-243 treatment. Together, these results confirm that the FLAG-ISG15 5xmutant can pass through both the ubiquitin and ISG15 assembly machineries.

(6) Most gels need to be run together to be able to compare the results. For example, in figure 5a and b, unless run together, it is not possible to really claim that the rates of UBE2L3 and UBE2L6 are comparable. In this figure in particular, as presented, it actually looks like UBA1-mediated UBE2L3 charging with ISG15 5xmut is enhanced compared to the

other gels. Other experiments should also be run together: Supp fig 9d, Supp fig 7f and g, panel in figure 4.

We appreciate this comment from reviewer 1 since it is an important point. However, we took this into account when designing and performing these experiments. For example, all assays directly compared in figure 4, were performed at the same time and resolved using SDS-PAGE gels that were poured in the same multi-gel casting system. Moreover, all gels were either scanned together or scanned using the same settings, preventing scan-to-scan variation. Similarly in Supplementary Fig. 9d (now Supplementary Fig. 10d in the updated manuscript) we used the same approach. However, Supplementary Fig. 7f and g (now combined to Supplementary Fig. 7f) were not run together in the original manuscript and therefore we have repeated this experiment with all 4 gels run and scanned together and have updated this figure accordingly. To further illustrate this point, we have included the raw uncropped images for each of these figures (please see Appendix). In the Appendix, a large box denotes the entire scanned area and within these images the cropped sections are shown with another box. The final figure is shown for comparison.

Relating to the particular case highlighted by the reviewer (i.e., Figure 5a, b), we have repeated this entire set of experiments to include UBA1-mediated UBE2L3 charging with ubiquitin. Overall, this reanalysis revealed similar findings. Slight differences in UBE1L-mediated UBE2L6 charging with wild-type ISG15 and ISG15^{5xmut} (Fig 5a) and, importantly, the ability of UBA1-mediated UBE2L3 charging with the 5xmutant (Fig 5b). Regarding the comparison of rates between UBE2L3 and UBE2L6 charging in Fig. 5a & b, we agree that our initial interpretation that the rates were comparable has the potential to cause confusion. However, our intention was to highlight that the levels of UBE2L3 charging with the ISG15^{5xmut} were similar to what we would expect for E2 enzymes functioning with their cognate Ub/Ubl and E1, thereby emphasising the successful identification of specificity determinants. A result that is further supported by screening the E2 panel and the transfection assays in the next figure (Fig. 6 in the updated manuscript). To avoid confusion, we have further improved the text of this section in the updated manuscript.

(7) In figure 5 the authors should also compare the UBA1-mediated UBE2L3 in when interacting with Ub.

As suggested by the reviewer, we have included this comparison in the updated version of the manuscript (see point 6 above).

(8) Some experiments lack positive or negative controls, figure 4h for example.

We appreciated this concern from reviewer #1 however this is not the case. As discussed above in point 6, all assays that are directly compared were performed and scanned at the same time. Therefore, in Figure 4g top left panel (UBE1L charging with wild-type ISG15) serves as a positive control, while in the top right panel (UBA1 charging with wild-type ISG15) serves as a negative control. We now realise that the figure layout may have caused this confusion, and we should not have divided this experiment into different sub-figures (i.e., g & h). In the resubmitted version of the manuscript, this experiment is no longer divided, and the data is shown as Figure 4g (see figure below).

Raw image from Figure 4g (formerly Fig. 4g &h)

(a) Supplemental Figure 8a left panel should include UBE2L6 as a control

As discussed above, all assays that are directly compared were performed at the same time, run together, and scanned together or scanned using identical settings. This includes Supplementary Fig. 8a (see below). Therefore, the second non-reducing gel which contains UBE2L6 serves as this control.

Raw image from Supplementary Figure 8a

Reviewer #2 (Remarks to the Author):

The manuscript by Wallace et al. reported the cryo-EM structures of UBE1L in complex with UBE2L6 and activated ISG15 which provides insights into the initial catalytic steps of the ISG15 conjugation pathway. They trapped a stable, yet functional complex of cross-linked UBE1L-UBE2L6 along with adenylated ISG15. Structural-based mutagenesis and biochemical experiments are appropriately performed and are supportive of the importance of the catalytically relevant residues and confirms the molecular basis of the UBE1L specificity for ISG15 and UBE2L6. I think this manuscript should be published, I only have minor comments which I listed here.

We would like to thank reviewer #2 for their positive assessment of our work and for highlighting our extensive biochemical and mutagenesis analysis on the specificity of the ISG15 transfer cascade. We have now included further experimental data based on the thoughtful comments of this reviewer which we feel has further strengthened our

manuscript. By including this new data and by addressing the comments below, we hope reviewer #2 will continue to be supportive of our manuscript.

Comments:

1. While testing the effect of viral effectors, authors have used Influenza WT NS1B and SARSCoV2 PLPpro activity dead mutant in the UBE1L self-conjugation reaction. I do not really understand the reason behind testing the PLPpro activity dead mutant in this reaction since they began testing the viral effectors which should be WT. Does the WT PLPpro inhibit UBE1L activation?

Our rationale behind using inactive PLpro (Cys111Ala) was to use this as a tool to study the importance of distinct ISG15 domains in the ISG15 pathway. Since mutation of the active site cysteine of DUBs have been reported to increase their affinity for their Ub/Ubl substrates (Morrow et al., 2018), this provided us with a high affinity ISG15 binder that recognises both the ISG15 N-lobe and C-lobe. By comparison NS1B interacts with ISG15 primarily through its N-lobe. Therefore, these reagents would allow us to study the importance of different ISG15 domains during UBE1L charging. As discussed in the manuscript, we predicted this interaction would inhibit UBE1L charging, while NS1B would not. Indeed, this is the case.

With regards to WT PLpro, we have now tested its ability to bind ISG15 and confirmed that as expected it does bind, however we have not been able to show that WT PLpro inhibits UBE1L charging. This is likely due to its reduced affinity for ISG15. As discussed above the aim of this study was to use these viral ISG15 binders to study the importance of the different ISG15 domains, which we achieved using inactive PLpro. In the resubmitted manuscript, we have updated the text in this section to emphasise this point, thereby preventing any over-interpretation our data.

2. Authors nicely show how ISG15 patch mutants can be made UBA1 compatible. On a similar note, from the paper it does seem like a natural step to prepare Ube2L6 specific mutations to make it more like Ube2L3 and test if it then becomes compatible with UBA1 or there are other specificity determinants.

In the original manuscript, we describe a UBE2L6 mutant with three UBE2L3 residues (i.e., M5R, V8M, M123A) which reduces the charging of UBE2L6 with ISG15 (Supplementary Fig. 6e, f; Supplementary Fig. 10 b-d). As suggested by reviewer 2, we have tested this UBE2L6 mutant for enhanced activity with the ubiquitin conjugation machinery. In these in vitro assembly reactions, we compared UBE2L3, UBE2L6, and UBE2L6^{M5R/V8M/M123A}. Notably, UBE2L6 can function with the Ub E1 (Kumar et al., 1997; Zhao et al., 2004) and we also observe this in our assays. However, UBE2L6^{M5R/V8M/M123A} showed increased ubiquitination levels, confirming that these residues increase the compatibility of UBE2L6 with the ubiquitin machinery (see panel below). We have included this experiment as Supplementary Fig. 10f. Also see point 4 from Reviewer #1 above.

Minor comments:

1. In the line 200-201, it is stated “Another potential interesting contact site surrounds of UBE2L6, which based on cryo-EM density contacts the SCCH domain” but a figure for this is missing?

This figure has been included in the updated version of the manuscript (see Supplementary Fig. 6f). While cryo-EM density allowed us to identify a potential E1-E2 interaction site, it was not sufficient to model the side chain of Met123. To determine if this interaction was important for Ub/Ubl E1-E2 transfer, we performed comparative biochemical assays with a mutation at this site (i.e., Met123Ala; please see Supplementary Fig. 10d-f).

2. Line no. 108- authors should mention that they used C599A mutant of UBE1L

This has been included in the updated version of the manuscript.

3. Line no. (175-177): It would be informative to have a structure figure with overlapping AMP orientation and identical active site residues

In the updated version of the manuscript, we have included an overlay of these structures, which highlights a similar AMP orientation and key active site residues. Additionally, we have included a sequence alignment comparing the adenylation domain active site residues of UBA1 and UBE1L (see Supplementary Fig. 5f).

4. Line 188-189: “In particular, ISG15’s Trp123 and P130 create a unique hydrophobic patch adjacent to Ub’s Ile44 patch and forms additional interactions with the hydrophobic beta-sheet.” It would be better if the authors also mention the interacting residues in UBE1L.

This has been included in the updated version of the manuscript.

5. Line 190-191: “Importantly, these hydrophobic contacts do not exist in the UBA1-ubiquitin complex but are instead replaced with ionic interactions.” Which residues are involved in these ionic interactions? In the suppl fig. 5f, we can only see the residues on Ub (R42 and Q49), but not on the UBA1.

Thank you for this comment. Supplementary Fig. 5f (now Supplementary Fig. 5g) has been updated to show this interaction.

6. Line No.- 195-196: “Modelling of a phylogenetically related ubiquitin E2, UBE2L3, onto UBE2L6 helped provide insight into this specificity” It would be informative to mention the sequence similarity.

This has been included in the updated version of the manuscript.

7. The y-axis is not labeled in suppl fig 3; fsc curve

This has been included in the updated version of the manuscript.

8. Supp Fig 9c- the catalytic Cys is Cys86 (seq alignment figure 9b), but in the structure figure 9c it is labeled as C85.

This has been corrected. Thank you.

Appendix

Uncropped image from Figure 4c

- Both gels run and imaged together.

Uncropped image from Figure 4d (see also Supplementary Fig. 7d)

- All reactions run on same gel.

Uncropped image from Figure 4g (Formerly Figure 4g & h)

g

- All gels were run and imaged together.

Uncropped image from Figure 5a & b

- All gels were run and imaged together.

Uncropped image from Supplementary Figure 7f (Formerly Supplementary Figure 7f & g)

Uncropped image from Supplementary Fig. 10d (Formerly Supplementary Fig. 9d)

- All gels were run and imaged together.

REVIEWERS' COMMENTS

Reviewer #1 (Remarks to the Author):

Authors have addressed all points raised.